# A Switch from Cell-Associated to Soluble PDGF-B Protects against Atherosclerosis, despite Driving Extramedullary Hematopoiesis

**DOI:** 10.3390/cells10071746

**Published:** 2021-07-10

**Authors:** Renée J. H. A. Tillie, Thomas L. Theelen, Kim van Kuijk, Lieve Temmerman, Jenny de Bruijn, Marion Gijbels, Christer Betsholtz, Erik A. L. Biessen, Judith C. Sluimer

**Affiliations:** 1Department of Pathology, Cardiovascular Research Institute Maastricht (CARIM), Maastricht University Medical Center, 6229 HX Maastricht, The Netherlands; renee.tillie@maastrichtuniversity.nl (R.J.H.A.T.); t.theelen@outlook.com (T.L.T.); k.vankuijk@maastrichtuniversity.nl (K.v.K.); lieve.temmerman@maastrichtuniversity.nl (L.T.); jennydebruijn@live.nl (J.d.B.); m.gijbels@maastrichtuniversity.nl (M.G.); erik.biessen@mumc.nl (E.A.L.B.); 2Department of Pathology, GROW-School for Oncology and Developmental Biology, Maastricht University Medical Center, 6229 HX Maastricht, The Netherlands; 3Department of Medical Biochemistry, Experimental Vascular Biology, Amsterdam UMC, University of Amsterdam, 1105 AZ Amsterdam, The Netherlands; 4Rudbeck Laboratory, Department of Immunology, Genetics and Pathology, Uppsala University, 751 85 Uppsala, Sweden; christer.betsholtz@igp.uu.se; 5Institute for Molecular Cardiovascular Research (IMCAR), RWTH Aachen University, 52074 Aachen, Germany; 6BHF Centre for Cardiovascular Sciences (CVS), University of Edinburgh, Edinburgh EH16 4TJ, UK

**Keywords:** PDGF-B, atherosclerosis, plaque stability, fibrosis, hematopoiesis

## Abstract

Platelet-derived growth factor B (PDGF-B) is a mitogenic, migratory and survival factor. Cell-associated PDGF-B recruits stabilizing pericytes towards blood vessels through retention in extracellular matrix. We hypothesized that the genetic ablation of cell-associated PDGF-B by retention motif deletion would reduce the local availability of PDGF-B, resulting in microvascular pericyte loss, microvascular permeability and exacerbated atherosclerosis. Therefore, *Ldlr*^-/-^*Pdgfb^ret^*^/*ret*^ mice were fed a high cholesterol diet. Although plaque size was increased in the aortic root of *Pdgfb^ret^*^/*ret*^ mice, microvessel density and intraplaque hemorrhage were unexpectedly unaffected. Plaque macrophage content was reduced, which is likely attributable to increased apoptosis, as judged by increased TUNEL+ cells in *Pdgfb^ret^*^/*ret*^ plaques (2.1-fold) and increased *Pdgfb^ret^*^/*ret*^ macrophage apoptosis upon 7-ketocholesterol or oxidized LDL incubation in vitro. Moreover, *Pdgfb^ret^*^/*ret*^ plaque collagen content increased independent of mesenchymal cell density. The decreased macrophage matrix metalloproteinase activity could partly explain *Pdgfb^ret^*^/*ret*^ collagen content. In addition to the beneficial vascular effects, we observed reduced body weight gain related to smaller fat deposition in *Pdgfb^ret^*^/*ret*^ liver and adipose tissue. While dampening plaque inflammation, *Pdgfb^ret^*^/*ret*^ paradoxically induced systemic leukocytosis. The increased incorporation of 5-ethynyl-2′-deoxyuridine indicated increased extramedullary hematopoiesis and the increased proliferation of circulating leukocytes. We concluded that *Pdgfb^ret^*^/*ret*^ confers vascular and metabolic effects, which appeared to be protective against diet-induced cardiovascular burden. These effects were unrelated to arterial mesenchymal cell content or adventitial microvessel density and leakage. In contrast, the deletion drives splenic hematopoiesis and subsequent leukocytosis in hypercholesterolemia.

## 1. Introduction

The normal artery wall consists of three layers: the intima, the medial layer and the adventitia from luminal inside to outside, respectively [1]. The intima consists of a single layer of endothelial cells (ECs), whereas the media consists of smooth muscle cells (SMCs) embedded in the extracellular matrix (ECM) [1]. The adventitia harbors connective tissue, mesenchymal cells (MCs), immune cells and blood vessels amongst others [1]. Atherosclerosis is characterized by plaque accumulation in the subendothelial space of the intimal layer [1]. Despite cholesterol lowering treatment applied in 71% of cardiovascular patients, atherosclerosis remains a major cause of death in western society [2]. Plaque rupture and subsequent luminal thrombus formation can cause life-threatening complications [3]. The switch from plaques with stabilizing features, such as high mesenchymal cell density and resulting collagen accumulation and a thick fibrous cap, is triggered by the accumulation of immune cells, apoptosis and angiogenesis [4]. These processes degrade the matrix of the plaque and its fibrous cap, which would shield thrombogenic content from the arterial lumen, and this biomechanically weakens the fibrous cap and plaque [4]. Indeed, the formation of intra-plaque microvessels originating from adventitia has been identified as source of intra-plaque hemorrhage, i.e., the leakage of blood components such as erythrocytes and leukocytes into the plaque [5]. Hence, plaque and adventitial microvessels are thought to increase disease progression and severity [6]. Causality between leakage of intraplaque microvessels and plaque instability remains to be addressed. 

The important criteria in the association between intraplaque microvessels and disease severity are microvessel quantity and quality. A stable microvessel consists of a single endothelial layer resting on a basement membrane and pericytes that cover the ECs to provide stability [6]. Thus, microvessel quality is defined by healthy EC morphology, intact endothelial junctions and, especially, the presence of surrounding pericytes [7]. Microvessels in ruptured human coronary artery plaques generally present with endothelial abnormalities and the absence of stabilizing pericytes [8]. Platelet-derived growth factor B (PDGF-B) has been identified as an important factor for intercellular communication between ECs and pericytes during early angiogenesis [9]. Sprouting ECs secrete PDGF-B, which binds to heparan sulfate proteoglycans in the ECM and on the cell surface through its retention motif, which is a short amino acid sequence in the protein’s C-terminus [10]. PDGF-B is thereby thought to form a growth factor gradient, guiding pericytes towards the ECs of the developing vessel [10]. Disruption of this gradient by deletion of the retention motif, resulting in a shorter isoform, was observed to induce pericyte loss and subsequently to increase microvascular leakage of the blood–brain barrier [11]. Vice versa, absence of pericyte coverage and vessel dysfunction can be restored by overexpressing PDGF-B [12]. Whole-body knockout (KO) of PDGF-B results in embryonic lethality caused by widespread bleedings [13], which is in line with excessive bleeding as a common side-effect of PDGF receptor tyrosine kinase inhibitors such as imatinib [14].

In addition to its role in microvessel stabilization, PDGF-B may have effects on other cell types involved in atherogenesis. PDGF-B exerts its functions on target cells by homo- or hetero-dimerization with PDGF-A, PDGF-B and subsequent binding to PDGF receptor-α (PDGFRα) or -β (PDGFRβ) on the cell surface [15]. PDGF-B is naturally produced and secreted with and without its C-terminal retention motif as a cell-associated (or ECM-associated) or soluble isoform, respectively [15]. Platelets produce the soluble PDGF-B isoform through intracellular proteolytic processing [16]. Furthermore, both isoforms are likely produced by vascular ECs and macrophages, amongst others [15]. Both isoforms have been shown to be biologically active [17]. PDGF-B is a mitogen that stimulates fibroblast and SMC proliferation and ECM formation [18,19]. Indeed, an in vivo graft model showed that keratinocyte expression of either soluble or cell-associated PDGF-B results in increased distal or proximal proliferation of dermal mesenchymal cells, respectively [17]. Moreover, immune cells have been shown to express PDGF-B, and hematopoietic KO of both PDGF-B forms resulted in a pro-inflammatory phenotype as it increased numbers of activated cluster of differentiation (CD) 4+ T cells in blood and caused monocyte accumulation in plaques of apolipoprotein E^-/-^ (ApoE^-/-^) mice [20]. This is in contrast to immunosuppressive effects of PDGFR tyrosine kinase inhibitors [14]. It remains unclear whether functions in atherogenesis are mediated by the soluble or cell-associated isoform of PDGF-B. Thus, we studied the effect of ablation of the cell-associated form of PDGF-B, by removing its retention motif and forcing a switch to soluble PDGF-B, on vascular cell function in atherosclerosis.

## 2. Materials and Methods

### 2.1. Experimental Animals

Animal experiments were conducted according to Dutch governmental and AHA guidelines [21] and approved by Dutch regulatory authorities. PDGF-B retention motif KO mice were kindly provided by Betsholtz [9]. The murine PDGF-B protein is 241 amino acids long. To delete the retention motif of PDGF-B, a premature translational stop codon was inserted into exon 6 of the *Pdgfb* gene (amino acid position 211). These mice were crossed with low density lipoprotein receptor KO (*Ldlr*^-/-^) mice from an in-house breeding colony, with the resulting mice referred to as *Pdgfb^ret^*^/*ret*^. The LDL receptor regulates the amount of circulating cholesterol. Knock out of this receptor results in increased cholesterol levels in the blood, making the mouse susceptible to developing atherosclerosis when fed a high cholesterol diet [22]. Compared to other murine atherosclerosis models, the lipoprotein profile in *Ldlr*^-/-^ mice most closely resembles the circulating lipoprotein profile in dyslipidemic humans [22]. PDGF-B wildtype (WT) *Ldlr*^-/-^ mice served as controls in this study (referred to as *Pdgfb^WT^*^/*WT*^). All mice were crossed back on a C57BL/6J *Ldlr*^-/-^ background at least nine times. Animals were housed in the laboratory animal facility of Maastricht University under standard conditions. Food and water were provided ad libitum during the experiment. Mice were housed in individually ventilated cages (GM500, Tecniplast, Buguggiate, Italy) with up to five animals per cage. Cages contained bedding (corncob, Technilab-BMI, Someren, The Netherlands) and cage enrichment and these were changed weekly, which reduced handling of the mice to one handling per week during non-intervention periods.

### 2.2. Atherosclerosis Induction, Treatments and Tissue Collection

At 10–25 weeks of age, male mice were fed a 10 week high cholesterol diet (HCD) containing 0.25% cholesterol (824171, Special Diet Services, Essex, UK, 15% cocoa butter, 10% maize starch, 20% casein, 40.5% sucrose and 5.95% cellulose) to induce atherosclerosis [22]. Only male mice were used to minimize the number of animals per experiment and thus to strictly adhere to the 3R principles (replacement, reduction and refinement). Two separate mouse experiments were performed. The first experiment exclusively entailed mice (*Pdgfb^WT^*^/*WT*^ *n* = 19, *Pdgfb^ret^*^/*ret*^ *n* = 10) that were fed the HCD. At the start of the second experiment (*Pdgfb^WT^*^/*WT*^ *n* = 16, *Pdgfb^ret^*^/*ret*^ *n* = 9), blood was collected from vena saphena to assess leukocyte counts on standard laboratory diet (R/M-H 25 kGy, Bio-Services, Uden, The Netherlands) with an automated hematology analyzer (XP-300, Sysmex, Norderstedt, Germany). Thereafter, mice received the 10 week HCD and were housed on metabolic cages once for 24 h to assess food intake. Furthermore, these mice were injected intraperitoneally with 25 mg/kg 5-ethynyl-2′-deoxyuridine (EdU, E10415, Invitrogen, Waltham, MA, USA) 24 h before sacrifice. All mice were euthanized by intraperitoneal pentobarbital injection (100 mg/kg, with 1:10 dilution before injection). 

During the second experiment, glucose concentration of whole blood from splenic artery was measured using a blood glucose meter (Contour TS, Bayer, Leverkusen, Germany). Furthermore, blood was collected from the right ventricle (experiment 1) or vena cava (experiment 2) in the presence of EDTA to assess leukocyte counts and other hematological parameters (XP-300, Sysmex), leukocyte EdU incorporation by flow cytometry and plasma cholesterol and triglyceride levels. In all mice, blood collection was followed by phosphate-buffered saline (PBS) perfusion via the left ventricle. Aortic root was excised and immediately embedded in optimum cutting temperature (OCT) compound (361603E, VWR chemicals, Radnor, PA, USA). Spleen, pancreas, liver, kidneys, interscapular brown adipose tissue, epididymal white adipose tissue (eWAT) and heart were dissected and weighed during the second experiment. Liver and eWAT were fixed in 4% paraformaldehyde (PFA) overnight and paraffin-embedded. Femur and tibia were dissected followed by the determination of length and weight of the right femur bone. Spleen, femur and tibia were used for leukocyte and/or progenitor cell flow cytometry combined with EdU incorporation detection. 

### 2.3. Histology and Immunohistochemistry

Serial cryosections (5 μm) from OCT compound embedded aortic roots were cut and stained with hematoxylin and eosin (HE) for blinded quantification of plaque size and necrotic core content in four sections of aortic root (at 25 µm intervals) using computerized morphometry (Leica QWin V3, Cambridge, UK). Necrotic core was defined as acellular and anuclear plaque areas rich in cholesterol clefts.

Atherosclerotic plaques were assessed for collagen (Sirius red area/plaque area, 09400, Polysciences, Warrington, PA, USA), iron (Perls Prussian blue), macrophage content (MOMA-2 area/plaque area), mesenchymal cell content (α smooth muscle actin (αSMA)+ area/plaque area, F3777, Sigma-Aldrich, Saint Louis, MO, USA), adventitial microvessel density (number of CD31+ microvessels/adventitial area, 550274, BD Biosciences, Franklin Lakes, NJ, USA) and PDGF-B (PDGF-B area/MOMA-2 on adjacent slides, ab23914, Abcam, Cambridge, UK). In short, slides were incubated with primary antibodies (MOMA-2, αSMA, CD31, PDGF-B) followed by peroxidase-based or alkaline-phosphatase-based immunohistochemical staining (see Appendix A for detailed information). For Perls Prussian blue staining, slides were incubated with a freshly prepared mix consisting of 1 part 2% HCl and 1 part 2% potassium hexacyanoferrate (II) to produce a reaction between ferric ions and potassium hexacyanoferrate (II), resulting in blue staining. Subsequent sensitization was performed using diaminobenzidine (K346811-2, Agilent, Santa Clara, CA, USA). For Sirius red staining, slides were incubated in 0.1% Sirius red in saturated picric acid and subsequently rinsed in 0.01 M HCl.

Quantifications were performed blinded using the Leica QWin software (V3, Cambridge, UK) by one observer with low intra-observer variability (<5%). Mean fibrous cap thickness was determined using ImageJ software Version 1.51S (as described in [23], Bethesda, MD, USA). Conversion of pictures to pseudofluorescent images was performed using the deconvoluting option in FIJI software Version 1.53c (Bethesda, MD, USA). 

Paraffin-embedded liver and eWAT samples were serially sectioned (4 and 7 μm, respectively) and HE-stained. Fat accumulation was scored blinded through visual analogue scores from 0 to 4.

### 2.4. Isolation and Culturing of Bone Marrow Cells

Femur and tibia of *Pdgfb^WT^*^/*WT*^ and *Pdgfb^ret^*^/*ret*^ mice on standard laboratory diet were dissected. Bone marrow cells were isolated by flushing bones with PBS. Single cell suspensions were obtained by passing cells through a 70 µm cell strainer. 

Bone marrow cells were cultured on non-tissue-culture-treated petri dishes in RPMI 1640 medium (72400047, Gibco, Waltham, MA, USA) or DMEM medium (31966021, Gibco) with 15% cell line L929-conditioned medium (LCM), 10% heat-inactivated fetal calf serum (FCS, FBS-12A, Capricorn Scientific, Ebsdorfergrund, Germany) and 1% Penicillin Streptomycin (P/S, 15070-063, Gibco). LCM was added to ensure differentiation of bone marrow-derived monocytes to macrophages (BMDMs). After 7 day differentiation, BMDMs were detached with lidocaine and generally plated onto non-tissue culture treated plates for various assays. Prior to the addition of stimuli, BMDMs were always allowed to attach overnight.

### 2.5. RNA and DNA Isolation

RNA and DNA were isolated using the TRIzol reagent (15596026, Thermo Fisher Scientific, Waltham, MA, USA) and subsequent chloroform phase separation which was performed following manufacturer’s protocol. Both DNA and RNA concentrations were determined with a NanoDrop 2000 (Thermo Fisher Scientific).

### 2.6. BMDM Genotype Confirmation

PDGF-B retention motif KO was assessed in BMDMs through DNA genotyping. A master mix containing REDExtract-N-Amp PCR ReadyMix (R4775, Sigma-Aldrich) and specific forward and reverse primers (10 µM, Appendix A) was added per DNA sample (≥100 ng) and separately for both the *Pdgfb^WT^* and *Pdgfb^ret^* gene. Subsequently, PCR was performed in a thermal cycler (MyCycler Thermal Cycler System, Bio-Rad, Hercules, CA, USA). An agarose gel was casted using agarose, 0.5×TBE buffer (tris-borate-EDTA buffer, 45 mM tris-borate 1 mM EDTA in dH_2_O) and SYBR Safe DNA Gel stain (1:35,000 dilution, S33102, Invitrogen). The resulting PCR samples and a DNA ladder (GeneRuler 100 bp Plus DNA ladder, SM0321, Thermo Fisher Scientific) were loaded and electrophoresis was performed for 30 min on 120 V (Powerpac 300, Bio-Rad). 

### 2.7. Reverse Transcription and Quantitative PCR

RNA was reverse transcribed to cDNA with the iScript cDNA synthesis kit following the manufacturer’s protocol (1708890, Bio-Rad). Subsequent quantitative polymerase chain reaction (qPCR) was performed using 10 ng cDNA, SYBR Green Supermix (1708885, Bio-Rad) and specific primer sets (Appendix A). The 18 Svedberg ribosomal RNA (18s rRNA) was used as a housekeeping gene to correct for different mRNA quantities between samples. Analysis was performed with CFX Manager Software Version 3.1 (Bio-Rad). 

### 2.8. ELISA PDGF-B

PDGF-B protein levels in BMDM conditioned medium were determined using ELISA (Quantikine ELISA, MBB00, R&D Systems, Minneapolis, MN, USA). The antibodies bind between amino acids 74 and 182 of the PDGF-B protein and, thus, not to the retention motif. ELISA was performed following the manufacturer’s protocol and read at 450 nm and 540 nm for wavelength correction with a SpectraMax M2 and SoftMax Pro Software Version 5 (Molecular Devices, San Jose, CA, USA).

### 2.9. High Content Analysis of BMDM Apoptosis and Cholesterol Uptake

BMDMs were plated onto Falcon 96-well tissue culture-treated imaging microplates (353219, Corning, Corning, NY, USA) for apoptosis and cholesterol uptake assays. BMDMs were stimulated with 7-ketocholesterol (C2394, Sigma-Aldrich, 50 µM), oxidized LDL (oxLDL, 50 µg/mL) and/or PDGF-B (SRP3229, Sigma-Aldrich, 140 pg/mL) for the apoptosis assay. After 22–24 h, BMDMs were incubated with Hoechst 33342 (4 µg/mL, B2261, Sigma-Aldrich) and washed with Annexin binding buffer (10 mM HEPES, 140 mM NaCl, 5 mM CaCl_2_, pH 7.4). Subsequently, cells were incubated with Annexin V-OG [24] (FP488, 2.6 µg/mL).

For cholesterol uptake, BMDMs were incubated with oxLDL (8 µg/mL) and Topfluor-labelled Cholesterol (2 µg/mL, 810255, Avanti Polar Lipids, Alabaster, AL, USA) for 3 h. Thereafter, BMDMs were incubated with Hoechst 33342 and washed.

Imaging was performed with a BD Pathway 855 High Content Analyzer (HCA, BD Biosciences) and 10-fold objective, taking 9 pictures/well. A digital segmentation mask for each cell based on nuclear Hoechst signal was created with BD Attovision Software Version 1.6. Automated analysis of output parameters for fluorescence probe intensity was performed and BD FACSDiva Software Version 6.1.2 was used for subsequent blinded analyses of apoptosis and lipid uptake by one observer with low intraobserver variability.

### 2.10. TUNEL Assay

In order to visualize apoptosis, a TUNEL (terminal deoxynucleotidyl transferase-mediated dUTP nick-end labeling) assay was performed on cryosectioned aortic roots (In Situ Cell Death Detection Kit, TMR Red, 12156792910, Roche, Basel, Switzerland) following the manufacturer’s protocol. Cell density, plaque area and number of apoptotic cells were determined blinded using QuPath Version 0.1.2 (Edinburgh, UK) and ImageJ Software Version 1.51S. 

### 2.11. BMDM Migration

A cross-shaped scratch was applied per well. Subsequently, pictures were taken (Leica DFC300 FX, Leica Microsystems, Wetzlar, Germany) of four fixed positions in each well at several time points (0, 0.5, 1 and 2 h) with a 10× objective (Leica DM IL microscope, Leica Microsystems). Migration over time was assessed with ImageJ Software version 1.51S. 

### 2.12. BMDM Proliferation

BMDMs were plated onto an E-plate view 96 (Acea, Roche), which was placed into a real-time cell analysis (RTCA) SP station (Acea, Roche) in an incubator on 37 °C with 5% CO_2_. Subsequent proliferation of BMDMs measured as change in electrical impedance was assessed with xCELLigence RTCA (Acea, Roche) and analyzed with RTCA Software 1.2. 

### 2.13. BMDM Matrix Metalloproteinase (MMP) Activity

BMDMs were lysed with 1% Triton X-100 in PBS and OmniMMP Fluorogenic Substrate (400 µM, BML-P126-0001, Enzo Life Sciences, Farmingdale, NY, USA) was added in the 1× Omni-buffer (50 mM HEPES, 10 mM CaCl_2_ in dH_2_O, pH 7.0). Fluorescence was measured at 2 min intervals from 0–300 min at an excitation of 320 nm and emission of 405 nm with a SpectraMax M2 and SoftMax Pro Software Version 5 (Molecular Devices).

### 2.14. Absolute Circulating Leukocyte Counts by Flow Cytometry 

Flow cytometry was performed to quantify absolute leukocyte subsets in whole blood. Blood was added to BD Trucount Absolute Counting Tubes (340334) containing Fc receptor block (anti-CD16/32 antibody). Thereafter, an antibody cocktail was added (Appendix A) and erythrocytes were lysed with lysis buffer (8.4 g/L NH_4_Cl and 0.84 g/L NaHCO_3_ in dH_2_O, pH 7.4) prior to measurement. All flow cytometry samples were measured with a BD FACSCanto II and analyzed with BD FACSDiva Software (BD Biosciences).

### 2.15. Leukocyte and Progenitor Cell Click-iT EdU Detection and Flow Cytometry

Flow cytometry was performed to assess leukocytes in blood and spleen and progenitor cells in bone marrow and spleen. Whole blood was centrifuged (2100 rpm, 10 min, 4 °C) and plasma was stored at −80 °C until further use. The spleen was crushed through a 70 µm cell strainer (542070, Greiner Bio-One, Kremsmünster, Austria) to obtain a single cell suspension. Femur and tibia were flushed with PBS and bone marrow cells were passed through a 70 µm cell strainer. Erythrocytes in all samples were lysed with lysis buffer. For flow cytometry of leukocytes, Fc receptors were blocked and, hereafter, an antibody mix was added. For flow cytometry of progenitor cells, antibody mix was added without prior Fc receptor blocking (Appendix A). After antibody incubation, the Click-iT reaction with an Alexa Fluor 488-coupled azide was performed following the manufacturer’s protocol (Click-iT EdU Alexa Fluor 488 Flow Cytometry Assay Kit, C10420, Invitrogen). 

### 2.16. Statistical Analyses 

Graphs are presented as mean ± standard error of the mean (SEM). Results were statistically analyzed with GraphPad Prism Version 6 (GraphPad Software Inc., San Diego, CA, USA). ROUT outlier analysis was performed and any outliers were excluded. Subsequently, normality (Shapiro–Wilk) and equal variances (F-test) analyses and the corresponding parametric or non-parametric testing were performed. * *p* < 0.05, ** *p* < 0.01 and *** *p* < 0.001. 

## 3. Results

### 3.1. Increased Plaque Stability in Pdgfb^ret/ret^ Mice, Unaffected Adventitial Plaque Vessel Quantity and Leakage

Firstly, prior works on PDGF-B protein and mRNA expression in murine plaques and cell types involved therein were confirmed. PDGF-B immunoreactivity was present in macrophages and ECs, which is in line with mRNA expression in these cell types in vitro (Figure 1A,B). Gene expression in SMCs and fibroblasts in vitro was neglectably low. Similar to these observations in mice, the single-cell RNA sequencing dataset by Wirka et al. from human atherosclerotic coronary arteries showed that *PDGFB* was mainly expressed in macrophages and endothelial cells, albeit by a low percentage of cells. *PDGFA* was mainly expressed by mesenchymal cells in the human plaque (Appendix A [25,26]).

To investigate the effect of a switch from the cell-associated to the soluble form of PDGF-B on atherosclerosis, *Pdgfb^ret^*^/*ret*^ and *Pdgfb^WT^*^/*WT*^ mice were fed a HCD for 10 weeks (Figure 1C). Plaque size was significantly increased (+69%) in aortic roots from *Pdgfb^ret^*^/*ret*^ compared to *Pdgfb^WT^*^/*WT*^ mice, while the necrotic core content was unchanged (Figure 1D). This effect was independent of circulating cholesterol and triglyceride levels, which were comparable between *Pdgfb^ret^*^/*ret*^ and *Pdgfb^WT^*^/*WT*^ mice (Appendix A). 

To clarify the process underlying increased plaque growth, we studied the plaque phenotype in these mice. As PDGF-B^ret/ret^ KO causes microvessel leakage in the blood–brain barrier [11], the effect on angiogenesis was studied. However, no CD31 positive microvessels could be detected in either *Pdgfb^ret^*^/*ret*^ or *Pdgfb^WT^*^/*WT*^ plaques (Figure 1E). Due to the lack of intra-plaque vessels, adventitial plaque vessel quantity was assessed as a surrogate parameter, which is in line with the outside-in hypothesis on the role of the adventitia in atherogenesis. However, in contrast to our expectations, adventitial vessel quantity also did not differ between genotypes. In line with these observations, no differences in intra-plaque or adventitial iron residues, potentially originating from lysed erythrocytes after blood vessel leakage, were found in *Pdgfb^ret^*^/*ret*^ nor *Pdgfb^WT^*^/*WT*^ mice (Figure 1F). In contrast, iron-laden macrophages were histologically observed in *Pdgfb^ret^*^/*ret*^ versus *Pdgfb^WT^*^/*WT*^ livers (Appendix A), indicating that PDGF-B retention motif KO did cause the leakage of blood vessels in other organs. Furthermore, extensive ECM accumulation in *Pdgfb^ret^*^/*ret*^ glomeruli was observed, confirming previous observations of microvascular leakage [9] (Appendix A). 

On the other hand, collagen content and mean thickness of the fibrous cap were also increased in *Pdgfb^ret^*^/*ret*^ plaques (Figure 1G). Plaques were then analyzed for the presence of alpha-smooth muscle actin (αSMA)-positive mesenchymal cells (MCs) and MOMA-2 antigen-positive macrophages, as these cell types play an important role in the production and degradation of collagen in the atherosclerotic plaque, respectively [27,28]. Furthermore, PDGF-B is a known inducer of MC proliferation [19]. Plaque macrophage content was decreased by half in *Pdgfb^ret^*^/*ret*^ mice (Figure 1H), whereas αSMA-positive MC content was unaffected (Figure 1E). Our data so far suggested that despite an increase in *Pdgfb^ret^*^/*ret*^ plaque size, the stability of *Pdgfb^ret^*^/*ret*^ plaques is increased due to increased collagen content and fibrous cap thickness and decreased macrophage content. These observations are unrelated to changes in intra-plaque microvessel quantity or leakage.

### 3.2. Soluble PDGF-B Secretion and Pdgfb^ret/ret^ Macrophage Susceptibility to Apoptosis Increased

In order to further investigate the decreased *Pdgfb^ret^*^/*ret*^ plaque macrophage content and its possible association with increased plaque collagen content, we studied whether the *Pdgfb^ret^*^/*ret*^ macrophage function was affected.

First, we confirmed the *Pdgfb* genotype in murine bone marrow-derived macrophages (BMDMs) isolated from *Pdgfb^ret^*^/*ret*^ and *Pdgfb^WT^*^/*WT*^ bone marrow (Figure 2A). This confirmation was provided by DNA genotyping as the KO of the retention motif was conferred by the introduction of a translational stop codon into the *Pdgfb* gene. Moreover, the unavailability of specific antibodies directed against the murine PDGF-B retention motif prevented KO confirmation at the protein level. Total *Pdgfb*, *Pdgfrb* and *Pdgfra* mRNA expression levels in *Pdgfb^ret^*^/*ret*^ macrophages were unchanged (Figure 2B). As expected, based on impaired retention of the protein to heparan sulfate proteoglycans on the cell surface [9,30], ablation of the cell-associated form resulted in the increased secretion of soluble PDGF-B into medium by *Pdgfb^ret^*^/*ret*^ macrophages (Figure 2C). This PDGF-B secretion was assessed with an antibody that binds between amino acids 100 and 200 of the murine PDGF-B protein and thus not to the retention motif. This observation is in line with higher PDGF-B protein immunoreactivity per macrophage-positive plaque area (Figure 2D,E).

Thus, we investigated whether enhanced soluble PDGF-B secretion in *Pdgfb^ret^*^/*ret*^ mice results in changes in macrophage functions, such as apoptosis, lipid uptake, proliferation, migration and matrix metalloproteinase (MMP) activity explaining the fibrotic plaque phenotype and reduced macrophage content. 

Indeed, apoptosis upon incubation with cholesterol oxidation products 7-ketocholesterol (7-KC) or oxidized low-density lipoprotein (oxLDL) was significantly increased in *Pdgfb^ret^*^/*ret*^ compared to *Pdgfb^WT^*^/*WT*^ BMDMs in vitro (Figure 3A–D). Increased susceptibility to apoptosis was not caused by increased lipid uptake (Figure 3E). Likewise, pro-inflammatory cytokine secretion by *Pdgfb^ret^*^/*ret*^ BMDMs was unchanged (Appendix A). To confirm whether, specifically, increased levels of extracellular soluble PDGF-B stimulate macrophage apoptosis, we incubated C57BL/6J BMDMs with 7-KC and a similar PDGF-B concentration as previously established in *Pdgfb^ret^*^/*ret*^ BMDM medium (Figure 2C, 140 pg/mL). Indeed, apoptosis was significantly increased upon combinatorial stimulation with both 7-KC and soluble PDGF-B versus stimulation with only 7-KC (Figure 3F,G). In line with these observations and with increased PDGF-B protein immunoreactivity per macrophage-positive plaque area, in vivo assessment confirmed enhanced plaque apoptosis (Figure 3H,I) and unaffected pro-inflammatory cytokine levels in plasma (Appendix A). As basal BMDM migration and proliferation in vitro were unchanged (Figure 3J,K, Appendix A), enhanced apoptosis may be the underlying reason for reduced macrophage content.

Reduced macrophage content and thus lower net collagen degradation might partly explain the enhanced plaque collagen accumulation in *Pdgfb^ret^*^/*ret*^ versus *Pdgfb^WT^*^/*WT*^ mice. In addition, MMP activity was decreased in *Pdgfb^ret^*^/*ret*^ versus *Pdgfb^WT^*^/*WT*^ BMDMs (Figure 3L). Together, higher apoptotic rates in *Pdgfb^ret^*^/*ret*^ macrophages due to increased extracellular soluble PDGF-B might explain diminished plaque macrophage content. In parallel, lower levels of macrophages with reduced MMP activity may be partly responsible for larger and more fibrotic plaques. 

### 3.3. Differential Systemic Effects of Pdgfb^ret/ret^ on Body Weight and Circulating Immune Cells

In addition to local vascular effects, we observed systemic effects of *Pdgfb^ret^*^/*ret*^ on body weight and circulating immune cells upon hypercholesterolemia. Body weight gain after 10 weeks of HCD was less in *Pdgfb^ret^*^/*ret*^ mice (Appendix A), while 24 h food intake was unchanged between *Pdgfb^WT^*^/*WT*^ and *Pdgfb^ret^*^/*ret*^ mice (Appendix A). Lower body weight gain could likely be explained by decreased weight of liver and epididymal white adipose tissue (eWAT) in *Pdgfb^ret^*^/*ret*^ mice, which is in line with histological observations of decreased fat accumulation in *Pdgfb^ret^*^/*ret*^ versus *Pdgfb^WT^*^/*WT*^ liver and eWAT (Appendix A). Additionally, blood glucose levels were 45% lower in *Pdgfb^ret^*^/*ret*^ mice after the diet (Appendix A). Thus, *Pdgfb^ret^*^/*ret*^ seems to protect against an unfavorable diet-induced (cardio) metabolic phenotype. 

In contrast to beneficial vascular and metabolic effects, systemic inflammation was amplified in *Pdgfb^ret^*^/*ret*^ mice. Although immune cell counts were similar between *Pdgfb^ret^*^/*ret*^ and *Pdgfb^WT^*^/*WT*^ mice on a standard laboratory diet (Appendix A), a striking general leukocytosis was observed in hypercholesterolemia (Figure 4A–K). The increased leukocytosis in *Pdgfb^ret^*^/*ret*^ mice during hypercholesterolemia is not associated with changes in systemic inflammation since circulating levels of inflammatory cytokines remain unchanged between *Pdgfb^ret^*^/*ret*^ and *Pdgfb^WT^*^/*WT*^ mice (Appendix A). 

Increased circulating immune cells may result from the increased proliferation of progenitor cells in bone marrow or spleen and/or leukocyte proliferation in circulation and spleen. Thus, we investigated if enhanced proliferation was underlying leukocytosis. Leukocytosis was again observed in this second experiment, reconfirming the phenotype (Appendix A). EdU labeled similar fractions of bone marrow common myeloid progenitors (CMPs) and granulocyte and monocyte progenitors (GMPs) in *Pdgfb^ret^*^/*ret*^ and *Pdgfb^WT^*^/*WT*^ mice, suggesting unchanged progenitor proliferation (Figure 5A–C). However, extramedullary hematopoiesis was heightened, as shown by increased relative EdU-positive counts within the CMP, granulocyte and CD8+ T cell populations in the spleen (Figure 5D–F and Appendix A). In addition, the EdU-positive fraction of CD4+ and CD8+ T cells also increased in circulation (Figure 5G and Appendix A). In summary, increased proliferation largely drives extramedullary hematopoiesis and subsequent leukocytosis in *Pdgfb^ret^*^/*ret*^ during hypercholesterolemia. Overall, despite the amplification of the systemic inflammatory burden, *Pdgfb^ret^*^/*ret*^ prevents weight gain and lipid storage and supports the development of a fibrotic plaque phenotype in hypercholesterolemia.

## 4. Discussion

In our current study, we have disrupted PDGF-B’s ability to anchor to the ECM or cell surface by deleting its retention motif (*Pdgfb^ret^*^/*ret*^) and thus the cell-associated form. However, the soluble form of PDGF-B is, nonetheless, a biologically active protein [16,17]; it is produced and significantly more secreted by *Pdgfb^ret^*^/*ret*^ macrophages. The biological relevance of cell-associated and soluble PDGF-B is largely unknown. Here, we show the beneficial vascular and systemic effects of *Pdgfb^ret^*^/*ret*^ in hypercholesterolemia. Unexpectedly, plaque stability increased in *Pdgfb^ret^*^/*ret*^ mice, unrelated to intraplaque or adventitial microvessel quantity and leakage. Additionally, *Pdgfb^ret^*^/*ret*^ prevented body weight gain through decreased fat accumulation in liver and WAT. Contrary to beneficial local and systemic effects, we observed systemic leukocytosis in *Pdgfb^ret^*^/*ret*^ hypercholesterolemia, which is likely driven by increased extramedullary hematopoiesis.

Surprisingly, *Pdgfb^ret^*^/*ret*^ neither affected plaque or adventitial microvessel number nor leakage. It was reported previously that *Pdgfb^ret^*^/*ret*^ reduced retinal microvessel density [9], suggesting reduced angiogenesis in normocholesterolemic *Pdgfb^ret^*^/*ret*^ mice. In the current study, no intra-plaque vessels were found in hypercholesterolemic mice, as expected. Our results coincide with numerous studies that did not observe plaque angiogenesis in the widely used *Ldlr*^-/-^ or *ApoE*^-/-^ mouse models of atherosclerosis [31,32,33,34]. This may be related to scarcity of plaque neovascularization in mouse models or difficulties with the conventional detection of intraplaque microvessels through CD31 endothelial cell imaging [34]. Indeed, additional interventions or genetic alterations are generally required to induce neovascularization and hemorrhage within the murine plaques. Adventitial microvessels are often studied, instead, as these have also been associated with atherosclerosis initiation and progression [34]. However, in adventitia underlying the plaques, we also did not observe any changes in angiogenic density or leakage. Histological observations did show iron-laden macrophages and, thus, signs of blood vessel leakage in adult *Pdgfb^ret^*^/*ret*^ liver, possibly suggesting that PDGF-B remains an important factor for blood vessel stabilization even in hypercholesterolemia. However, this process seems to be organ-specific as it is dispensable for the permeability of arterial vasa vasorum. Therefore, we postulate that other factors such as vascular endothelial growth factors might be involved in the integrity of aortic intraplaque and adventitial microvessels in hypercholesterolemia [35]. 

Instead of affecting plaque or adventitial microvessel number and leakage, PDGF-B retention motif deletion showed protective local vascular and systemic effects. Although plaque size was increased, *Pdgfb^ret^*^/*ret*^ showed a more stable plaque phenotype as indicated by the increased collagen content and fibrous cap thickness and decreased macrophage content with similar MC content. While MC content was unaltered, PDGF-B is a well-established inducer of MC proliferation and has been associated with SMC migration from media to intima in atherosclerosis [36]. In line with our findings, normocholesterolemic *Pdgfb^ret^*^/*ret*^ mice without *Ldlr*^-/-^ background also increased collagen content without changing SMC numbers in the aortic media [37]. In contrast, in *ApoE*^-/-^ mice with loss of both cell-associated and soluble PDGF-B in hematopoietic cells, fibrous cap formation and MC accumulation were reduced and the plaque size was unaffected [38]. These results suggest that soluble PDGF-B is not responsible for MC content in early lesions, but does affect collagen accumulation in atherosclerotic plaque. 

The increased plaque collagen content may partly be explained by increased plaque macrophage apoptosis. This could be due to reduced survival signaling or increased apoptosis signaling. In favor of the former, in a vascular graft model with a knockout of both soluble and cell-associated PDGF-B in myeloid cells, Onwuka et al. also observed increased macrophage apoptosis and suggested an autocrine role for PDGF-B in macrophage maintenance [39]. Here, we show that enhanced secretion of soluble PDGF-B, in the absence of cell-associated PDGF-B, magnifies apoptosis induced by atherogenic stimuli such as 7-KC. Supplementation of soluble PDGF-B to BMDMs without genetic ablation of cell-associated PDGF-B also enhanced apoptosis. Thus, cell-associated PDGF-B might (partly) protect against apoptosis but, despite its presence, increased levels in soluble PDGF-B further stimulate apoptosis induced by cholesterol oxidation products. In this case, as previous studies mainly reported the protective effects of PDGF-B against apoptosis, the current study provides new insights regarding apoptosis-stimulating effects of soluble PDGF-B in atherosclerosis. In growth-arrested SMCs, soluble PDGF-B induced apoptosis through upregulation of B-cell leukemia/lymphoma (Bcl)-xs and downregulation of Bcl-2 and Bcl-xl gene expression [40]. This remains to be confirmed for macrophages.

In addition to macrophage apoptosis, increased *Pdgfb^ret^*^/*ret*^ plaque collagen content could also be explained by reduced macrophage MMP activity. Several studies support enhanced MMP activity, specifically the activity of MMP-2 and -9, in response to PDGF-B. MMP-9 secretion was increased in human macrophages after PDGF-B stimulation, although the isoform was unspecified [41]. Additionally, total PDGF-B overexpression is associated with MMP-2 and MMP-9 expression in murine liver [42]. Moreover, a study in ApoE^-/-^ mice reported decreased plaque MMP-2 and MMP-9 expression after injection with AG1296, which is a PDGFR inhibitor [43]. Together, these data suggest a stimulating effect of enhanced cell-associated PDGF-B on MMP activity and thus reduced local availability of cell-associated PDGF-B underlying reduced macrophage MMP activity. 

In addition to beneficial vascular effects, we observed systemic protection of *Pdgfb^ret^*^/*ret*^ against adiposity. Similar results were reported in a model of tamoxifen-inducible systemic PDGFRβ ablation in diet-induced obesity [44]. Our observations seem linked to previous reports of increased insulin signaling in *Pdgfb^ret^*^/*ret*^ liver that is caused by increased vascular permeability [45] and are thus in line with reduced local PDGF-B availability. 

Contrary to decreased *Pdgfb^ret^*^/*ret*^ plaque inflammation, systemic leukocytosis was a prominent feature of *Pdgfb^ret^*^/*ret*^ mice, with expansion of almost all subsets in the circulation. This effect was only observed in hypercholesterolemia and not in *Pdgfb^ret^*^/*ret*^ mice on standard laboratory diet. This is in line with observations that hematopoietic PDGF-B was not essential for basal hematopoiesis in normolipidemia, although the spleen was not studied [46]. Here, leukocytosis was likely caused by increased extramedullary hematopoiesis. Xue et al. overexpressed PDGF-B in subcutaneous tumors and reported heightened numbers of granulocyte-macrophage colony-forming units after isolation and stimulation of splenic progenitors in culture [47]. Indeed, we observed a trend in the percentage of GMPs (Appendix A) in *Pdgfb^ret^*^/*ret*^ spleen. Additionally, we report increased proliferation of CMPs in *Pdgfb^ret^*^/*ret*^ spleen in vivo. Thus, we show that extramedullary hematopoiesis is heightened due to increased progenitor proliferation and that PDGF-B already acts on CMPs to stimulate GMPs downstream, which has not been previously reported to the best of our knowledge. Moreover, Xue et al. reported PDGF-B-induced erythropoietin expression in PDGFRβ-expressing stromal cells as a cause of increased extramedullary hematopoiesis [47]. Thus, as the cell-associated PDGF-B is absent, the soluble PDGF-B isoform and possibly its increased secretion might be the main stimulator of splenic hematopoiesis in our model. We speculate that, in humans, the blockage of soluble PDGF-B by PDGFR inhibitors possibly underlies their immune suppressing effects [14]. Overall, PDGFR inhibition is associated with enhanced cardiovascular risk [48] and more insight into the PDGF-B isoforms may inform the design of future generation inhibitors. 

In the future, antibodies that specifically bind to the murine PDGF-B retention motif, which are currently unavailable, are warranted to further study the effects and affinity of PDGF-B isoforms and to clarify if underlying mechanisms are related to reduced bioavailability, altered (hetero)dimerization and/or signaling in PDGF-B-responsive and PDGF-B-producing cells. 

In conclusion, *Pdgfb^ret^*^/*ret*^ has a dual effect in hypercholesterolemia as it results in more stable plaques and protects against an unfavorable diet-induced metabolic phenotype on one hand; on the other hand, it stimulates an immune response by increasing extramedullary hematopoiesis. Thus, the current study warrants further investigation of downstream pathways to isolate beneficial and detrimental effects of the PDGF-B isoforms and underlying mechanisms. Furthermore, integrity and density of intraplaque or adventitial microvessels seem to be independent of cell-associated PDGF-B. 

## 5. Conclusions

The current study shows that *Pdgfb^ret^*^/*ret*^ confers protective vascular effects in the atherosclerotic plaque by affecting macrophage viability, through increased secretion of soluble PDGF-B, and MMP activity. Unexpectedly, *Pdgfb^ret^*^/*ret*^ does not affect intraplaque or adventitial microvessel density and leakage, suggesting that other factors are responsible for the stabilization of microvessels in the aortic adventitia. Furthermore, *Pdgfb^ret^*^/*ret*^ seems to protect against an unfavorable diet-induced metabolic phenotype, as indicated by decreased fat accumulation in liver and adipose tissues. In contrast to reduced *Pdgfb^ret^*^/*ret*^ plaque inflammation, general leukocytosis is observed due to hypercholesterolemia that is driven by increased extramedullary hematopoiesis. 

## Figures and Tables

**Figure 1 cells-10-01746-f001:**
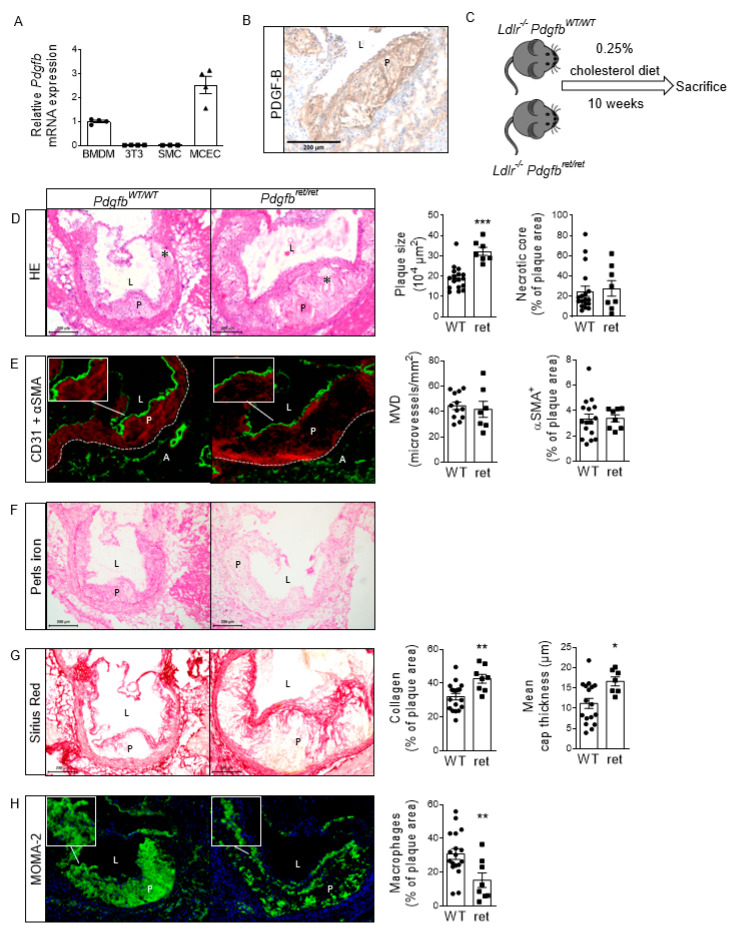
Plaque characteristics show larger but more stable *Pdgfb^ret^*^/*ret*^ plaques. (**A**) *Pdgfb* mRNA expression in mouse cardiac endothelial cells (MCECs [29]), NIH/3T3 fibroblasts and SMCs relative to BMDMs (*n* = 3-4). (**B**) PDGF-B immunoreactivity in *Ldlr*^-/-^ aortic root lesions. (**C**) Setup of mouse experiment using *Ldlr*^-/-^*Pdgfb^WT^*^/*WT*^ (*n* = 19) and *Ldlr*^-/-^*Pdgfb^ret^*^/*ret*^ (*n* = 10) mice. Representative photomicrographs of (**D**) HE, (**E**) CD31 + αSMA (green and red, respectively, pseudofluorescence), (**F**) Perls iron, (**G**) Sirius Red and (**H**) MOMA-2 (green, pseudofluorescence) staining *Pdgfb^WT^*^/*WT*^ (*n* = 12-18) and *Pdgfb^ret^*^/*ret*^ (*n* = 7-8) aortic root lesions and the corresponding quantifications. Nuclear staining in MOMA-2 staining of aortic roots was performed with hematoxylin (blue, pseudofluorescence). MVD; adventitial microvessel density. * in photomicrographs indicates necrotic core. P; plaque, L; lumen, A; adventitia. Graphs represent mean ± SEM. * *p* < 0.05, ** *p* < 0.01, *** *p* < 0.001. Scale bars 200 µm. Data were tested for normality (Shapiro-Wilk) and equal variances (F-test). Variables that did or did not pass were analyzed using Student’s *t*-test or the Mann-Whitney U test, respectively.

**Figure 2 cells-10-01746-f002:**
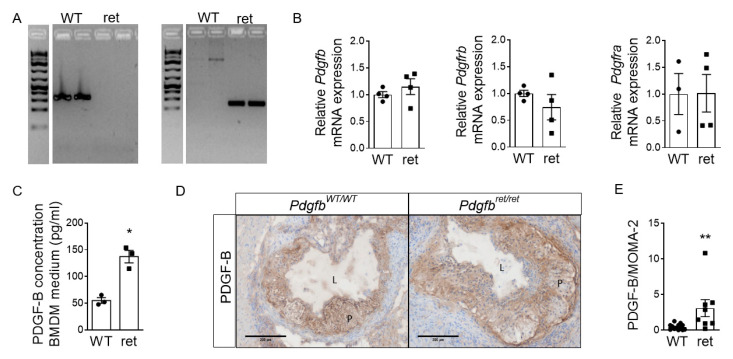
Increased secretion of soluble PDGF-B by *Pdgfb^ret^*^/*ret*^ BMDMs. (**A**) Gel imaging after DNA genotyping. PCR product *Pdgfb^WT^* 340 bp (left) and *Pdgfb^ret^* 212 bp (right). (**B**) *Pdgfb*, *Pdgfrb* and *Pdgfra* mRNA expression in *Pdgfb^ret^*^/*ret*^ relative to *Pdgfb^WT^*^/*WT*^ BMDMs (*n* = 4). (**C**) PDGF-B concentration in *Pdgfb^WT^*^/*WT*^ and *Pdgfb^ret^*^/*ret*^ BMDM-derived medium as assessed by ELISA (*n* = 3). (**D**) Representative photomicrographs of PDGF-B staining in *Pdgfb^WT^*^/*WT*^ and *Pdgfb^ret^*^/*ret*^ aortic root lesions. P; plaque, L; lumen. (**E**) Quantification of total PDGF-B plaque area relative to the total MOMA-2 plaque area in the adjacent sections of *Pdgfb^WT^*^/*WT*^ (*n* = 13) and *Pdgfb^ret^*^/*ret*^ (*n* = 8) aortic root lesions. Graphs represent mean ± SEM. * *p* < 0.05, ** *p* < 0.01. Scale bars 200 µm. Data were tested for normality (Shapiro-Wilk) and equal variances (F-test). Variables that did or did not pass were analyzed using Student’s *t*-test or the Mann-Whitney U test, respectively. Data in C were transformed to ranks followed by a Student’s *t*-test.

**Figure 3 cells-10-01746-f003:**
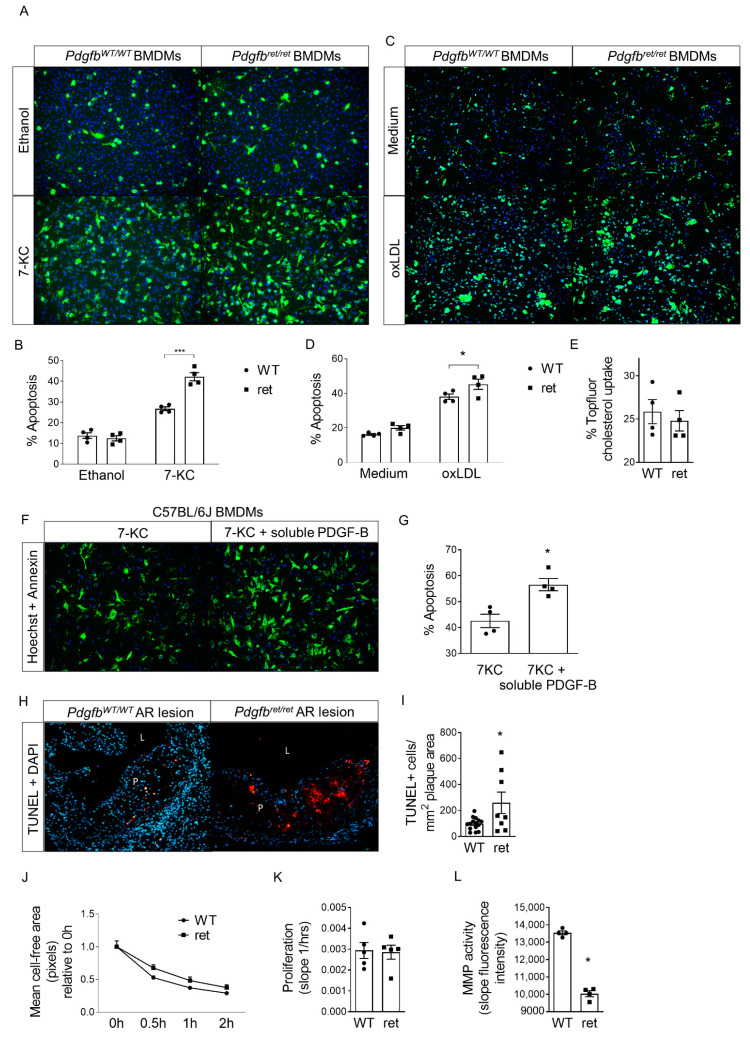
Increased apoptosis upon incubation with atherosclerosis-relevant stimuli and decreased MMP activity in *Pdgfb^ret^*^/*ret*^ BMDMs. All experiments in (**A**–**E**) and (**J**–**L**) were performed with *Pdgfb^WT^*^/*WT*^ and *Pdgfb^ret^*^/*ret*^ BMDMs. Representative photomicrographs of BMDM apoptosis stained with Annexin A5 (FP488, green) and Hoechst 33342 (blue) after 24 h incubation with (**A**) ethanol or 7-ketocholesterol (7-KC) or (**C**) medium or oxidized LDL (oxLDL), with corresponding quantification (*n* = 4) in (**B**) and (**D**). (**E**) Quantification of Topfluor-labeled cholesterol uptake by BMDMs after 3 h (*n* = 4). (**F**) Representative photomicrographs of C57BL/6J BMDM apoptosis stained with Annexin A5 (FP488, green) and Hoechst 33342 (blue) after 24 h incubation with 7-ketocholesterol and soluble PDGF-B, with corresponding quantification (*n* = 4) in (**G**). (**H**) Representative photomicrographs of TUNEL (red) and DAPI (blue) staining in *Pdgfb^WT^*^/*WT*^ (*n* = 15) and *Pdgfb^ret^*^/*ret*^ (*n* = 8) aortic root (AR) lesions and (**I**) quantification of TUNEL+ cells. P; plaque, L; lumen. (**J**) BMDM migration defined as mean cell-free area over time after scratch infliction (*n* = 4). (**K**) Mean BMDM proliferation measured as change in electrical impedance over time (*n* = 5). (**L**) Quantification of BMDM MMP activity with OmniMMP Fluorogenic Substrate (*n* = 4). Graphs represent mean ± SEM. * *p* < 0.05, ** *p* < 0.01. Data were tested for normality (Shapiro-Wilk) and equal variances (F-test). Variables that did or did not pass were analyzed using Student’s *t*-test or the Mann-Whitney U test, respectively. (**B**,**D**,**J**) were analyzed using two-way ANOVA and two-way repeated measures ANOVA, respectively, including Bonferroni’s multiple comparisons test.

**Figure 4 cells-10-01746-f004:**
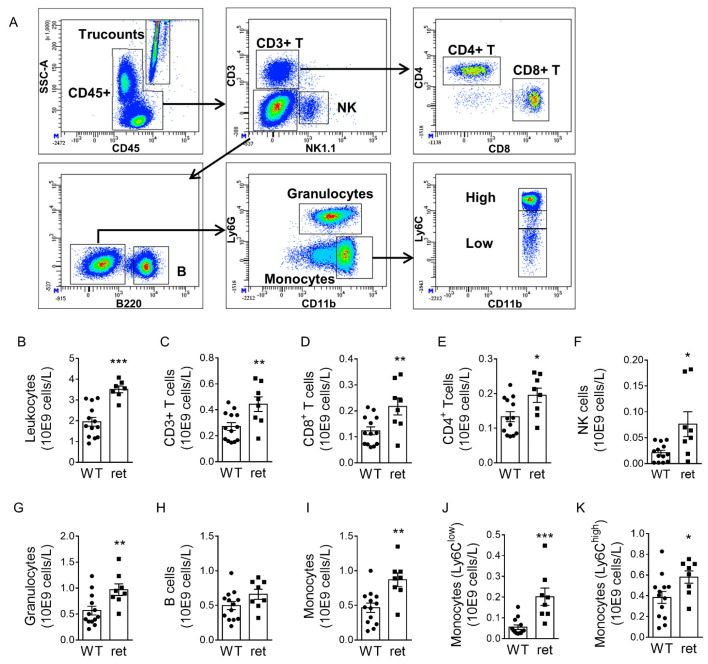
General leukocytosis in *Pdgfb^ret^*^/*ret*^ mice. (**A**) Flow cytometry gating strategy to assess absolute circulating leukocyte counts. (**B**–**K**) Absolute numbers of CD45+ leukocytes; CD3+, CD8+ and CD4+ T cells; NK (natural killer) cells; granulocytes; B cells and Ly6C^low^ and Ly6C^high^ monocytes in *Pdgfb^WT^*^/*WT*^ (*n* = 13) and *Pdgfb^ret^*^/*ret*^ (*n* = 8) blood. Graphs represent mean ± SEM. * *p* < 0.05, ** *p* < 0.01, *** *p* < 0.001. Data were tested for normality (Shapiro-Wilk) and equal variances (F-test). Variables that did or did not pass were analyzed using Student’s *t*-test or the Mann-Whitney U test, respectively.

**Figure 5 cells-10-01746-f005:**
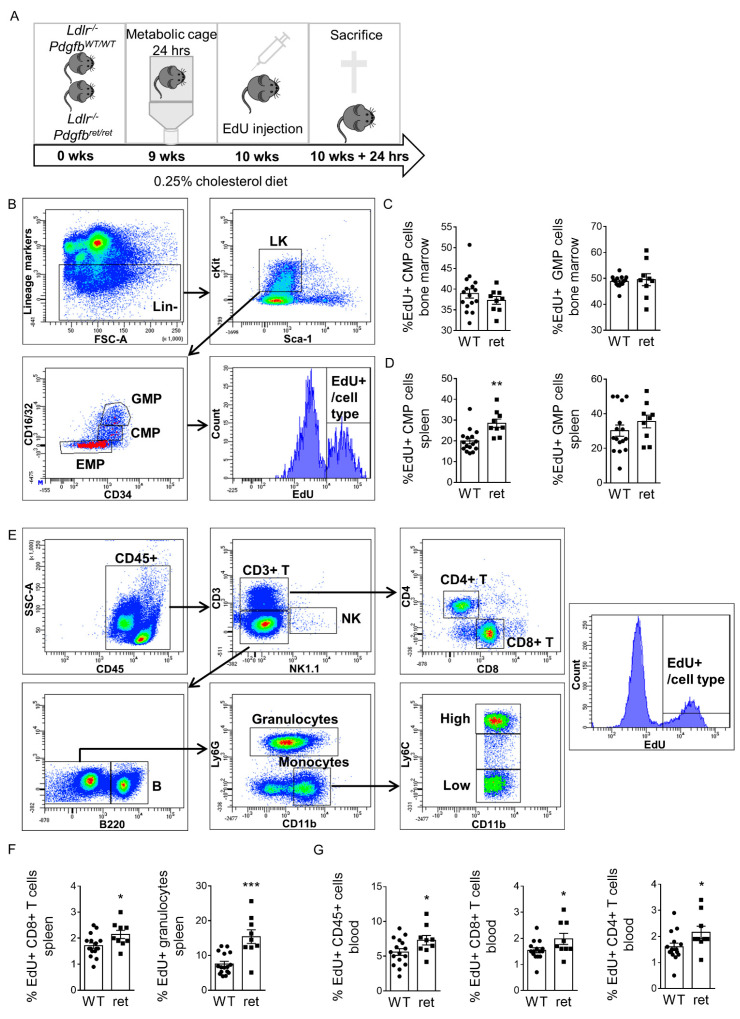
Increased extramedullary hematopoiesis and proliferation of leukocytes in *Pdgfb^ret^*^/*ret*^ mice. (**A**) Setup of second mouse experiment using *Ldlr*^-/-^*Pdgfb^WT^*^/*WT*^ (*n* = 16) and *Ldlr*^-/-^*Pdgfb^ret^*^/*ret*^ (*n* = 9) mice. (**B**) Flow cytometry gating strategy to assess hematopoietic progenitor cells and 5-ethynyl-2′-deoxyuridine (EdU) incorporation. (**C**) Percentage of EdU positive progenitor cells in *Pdgfb^WT^*^/*WT*^ (*n* = 16) and *Pdgfb^ret^*^/*ret*^ (*n* = 9) bone marrow and (**D**) spleen. (**E**) Flow cytometry gating strategy to assess leukocytes and EdU incorporation. (**F**) Percentage of EdU positive leukocytes in *Pdgfb^WT^*^/*WT*^ (*n* = 15-16) and *Pdgfb^ret^*^/*ret*^ (*n* = 9) spleen and (**G**) blood. Graphs represent mean ± SEM. * *p* < 0.05, ** *p* < 0.01, *** *p* < 0.001. Data were tested for normality (Shapiro-Wilk) and equal variances (F-test). Variables that did or did not pass were analyzed using Student’s *t*-test or the Mann-Whitney U test, respectively.

## Data Availability

The data that support the findings of this study are available from the corresponding author upon reasonable request.

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
