# Peer review of "A Switch from Cell-Associated to Soluble PDGF-B Protects against Atherosclerosis, despite Driving Extramedullary Hematopoiesis"

_cells, 2021, doi:10.3390/cells10071746_

Round 1
Reviewer 1 Report
The Authors in the present original article evaluated the beneficial vascular and systemic effects of Pdgfbret/ret in an hypercholesterolemic animal model. In particular, They observed that Pdgfbret/ret incresed plaque stability, prevented body weight gain through decreased fat accumulation in liver and white adipose tissue. In contrast to these beneficial vascular and systemic effects, Authors observed also systemic leukocytosis.
Good article. I think that the present manuscript has an interesting topic and it could be a starting point for a better understanding of the pathophysiological process(es) involved in atherosclerotic plaque formation and, interestingly, it provided important evidence about PDGF-B involvement in vascular and metabolic functions.
In my opinion, this manuscript has a clear message, the rationale for the choice of the experimental model as well as the technical approaches used are appropriate. The obtained results are fully described and discussed.
However, I have some mandatory comments:
- In all the text check that abbreviations are cited in full the first time whic are reported (such as at lines 54 or 126).
Introduction
- Some references that justify the sentences reported are missing (better if recent);
- briefly introduce vessel morphological and anatomical features.
Materials and Methods
- Line 100: specify the sex of the animals used. Sex differences may be observed?
- lines 100, 110: add, if possible, one or more references that justify the animal model and treatments used (or briefly discuss it);
- lines 164, 175, 181 and 234: specify if the all morphometrical analyses were done in blind and if the observers were two o three;
- lines 166-172: briefly describe the immunohistochemistry tecnique used and also specify the primary antibodies dilutions.
Results
- Figure 1: identify in the photomicrographs vessel lumen, ECs, macrophages and necrotic core. Furthermore, a zoom of macrophages immunopositivity should be useful for the readers (Fig. 1);
- in all the Results paragraph describe the data in the order in which were presented in the figures;
- in all the photomicrographs indicate the vessel lumen and the atherosclerotic plaque;
- Fig. 1E αSMA: the nuclei (DAPI) are not visible in the immunofluorescent photomicrographs. A zoom relative of CD31 immunopositivity may be useful for the readers (especially to better identify endothelial positivity);
- Fig. 2D: the photomicrographs are too light and the immunopositivity is not clearly visible.
Discussion
- Line 465: add other reference(s) that justify the sentence due to the Authors reported “Our results coincide with numerous studies …”;
- the Discussion paragraph is too long, please reduces it;
- a simple schematic graph that summarized the data obtain should be useful for the readers.
Author Response
Dear Reviewer 1,
Thank you for your suggestions and comments on our manuscript. Please find our response to your comments (and to the comments of reviewer 2) attached.
Kind regards,
Renée Tillie

Reviewer 2 Report
The article intituled “A switch from cell-associated to soluble PDGF-B protects against atherosclerosis, despite driving extramedullary hematopoiesis”, describes the effect of soluble PDGFb in plaque formation and stabilization.
The article is well written and contains interesting results and uses extensive methods to determine the role of PDGF b in atherosclerosis.
Specific comments:
English corrections – it’s better to use tibia and femur in line 183
Material and methods:
It would be easier to follow the results if the material and methods were in the same order, e.g., items 2.3 and 2.4 described the flow cytometry, and the results were presented in figures 4 and 5.
Just as a comment, some methods were added in the article, but the results are in the supplemental section, it could be better to add material and methods to the supplemental section and leave the comment of these results in the main article.
Results:
Is there a characterization of the BMDM using flow cytometry?
Images of the cell scratch assay could be added to the figure or to the supplemental section
In figure 2E, the authors present a ratio between MOMA-2 and PDGF but do not add the correspondent figure for that
In figure 3, there is a mixture of in vitro and in vivo experiments that makes the figure confusing, the authors could change the position of figure 3 H and I put it at the top and then compare the results in vivo and in vitro
Since the authors are comparing the in vivo and in vitro responses it could be very interesting to have an in vitro analysis of gelatinase activities of the tissue using for example a chromogenic substrate.
Discussion:
The authors could discuss why they consider plaque of the PDFG b ret more stable.
Author Response
Dear Reviewer 2,
Thank you for your suggestions and comments on our manuscript. Please find our response to your comments (and to the comments of reviewer 1) attached.
Kind regards,
Renée Tillie
